# Evaluation of Physiological and Morphological Traits for Improving Spring Wheat Adaptation to Terminal Heat Stress

**DOI:** 10.3390/plants10030455

**Published:** 2021-02-28

**Authors:** Hafeez ur Rehman, Absaar Tariq, Imran Ashraf, Mukhtar Ahmed, Adele Muscolo, Shahzad M. A. Basra, Matthew Reynolds

**Affiliations:** 1Department of Agronomy, University of Agriculture, Faisalabad 38000, Pakistan; hafeezcp@gmail.com (H.u.R.); arid192@gmail.com (A.T.); imran.ashraf@post.com (I.A.); shehzadbasra@gmail.com (S.M.A.B.); 2Department of Agricultural Research for Northern Sweden, Swedish University of Agricultural Sciences, 90183 Umeå, Sweden; 3Department of Agronomy, Pir Mehr Ali Shah, Arid Agriculture University, Rawalpindi 46300, Pakistan; 4Department of Agriculture, Mediterranea University, Feo di Vito, 89124 Reggio Calabria, Italy; amuscolo@unirc.it; 5International Maize and Wheat Improvement Center, El Batán, Texcoco CP 56130, Mexico; m.reynolds@cgiar.org

**Keywords:** canopy temperature, water soluble carbohydrates, heat stress, stay green, seed yield

## Abstract

Wheat crop experiences high temperature stress during flowering and grain-filling stages, which is termed as “terminal heat stress”. Characterizing genotypes for adaptive traits could increase their selection for better performance under terminal heat stress. The present study evaluated the morpho-physiological traits of two spring wheat cultivars (Millet-11, Punjab-11) and two advanced lines (V-07096, V-10110) exposed to terminal heat stress under late sowing. Early maturing Millet-11 was used as heat-tolerant control. Late sowing reduced spike length (13%), number of grains per spike (10%), 1000-grain weight (13%) and biological yield (15–20%) compared to timely sowing. Nonetheless, higher number of productive tillers per plant (19–20%) and grain yield (9%) were recorded under late sowing. Advanced lines and genotype Punjab-11 had delayed maturity and better agronomic performance than early maturing heat-tolerant Millet-11. Advanced lines expressed reduced canopy temperature during grain filling and high leaf chlorophyll *a* (20%) and *b* (71–125%) contents during anthesis under late sowing. All wheat genotypes expressed improved stem water-soluble carbohydrates under terminal heat stress that were highest for heat-tolerant Millet-11 genotype during anthesis. Improved grain yield was associated with the highest chlorophyll contents showing stay green characteristics with maintenance of high photosynthetic rates and cooler canopies under late sowing. The results revealed that advanced lines and Punjab-11 with heat adaptive traits could be promising source for further use in the selection of heat-tolerant wheat genotypes.

## 1. Introduction

Wheat (*Triticum aestivum* L.) is a staple crop and nourishes billions of people daily, while its productivity is significantly decreased under high temperature. Owing to global climate changes, wheat yields are expected to decline by 6% for each 1 °C increase in temperature. Therefore, gain yield of wheat crop must be increased by 60% until 2050 to fulfil the food demands of burgeoning global population [1]. According to the International Maize and Wheat Improvement Center (CIMMYT), world wheat producing regions are grouped into eight different mega environments (MEs) [2]. The ME1 and ME5 are highly productive irrigated environments in South Asia; however, wheat crop is exposed to terminal heat stress in these MEs [3]. South Asia includes India, Nepal, Pakistan and Bangladesh where wheat is sown late due to the delay in the harvesting of previous crops, i.e., rice and cotton in rice-wheat and cotton-wheat zones owing to delayed harvest of long maturing semi-dwarf rice varieties and delayed picking of cotton respectively [4,5]. The delay in wheat cultivation exposes the crop to high temperatures during flowering and grain-filling stages termed as “terminal heat stress” [6]. Nonetheless, wheat grown on ≥13.9 million ha in the rice-wheat cropping system of South Asia [1] including rice-wheat and cotton-wheat zones of Pakistan experience terminal heat stress [4,5].

Spring wheat in South Asia is sown in the months of November and December while harvested during April and May particularly in rice-wheat and cotton-wheat cropping zones. The timely planted wheat crop in these zones has temperature requirement of 12–22 °C during vegetative and reproductive growth periods particularly at anthesis and grain filling stages.

Late sown wheat experiences higher canopy temperature (>31 °C) during reproductive period adversely affecting many physiological processes resulting in shortened crop growth cycle and reduced yields [6]. For example, high temperature during anthesis reduces pollen viability by restricting embryo development and promotes anther sterility resulting in less grain numbers [7,8]. High temperature exposure after anthesis decreases grain filling rate which is associated with declined grain yield [7,8,9,10]. These decreases in grain filling rate and duration accelerate senescence with the loss of chlorophyll and limits assimilate supply towards developing grains [11]. 

Nonetheless, wheat plants evolve several physiological mechanisms to cope with terminal heat stress which include early maturity [3,6], stay green [11,12], reduced canopy temperature [13], accumulation of high stem water soluble carbohydrates [14] and high biomass accumulation [13] to translate assimilates into yield. For instance, early maturity as an adaptation strategy provides explanation for variation to physiological responses and grain yield under high temperature among wheat genotypes during reproductive period [3,6]. Stay green trait maintains high leaf chlorophyll contents that are associated with grain yield and its components under heat stress [11,12,15]. Likely, the loss in green area affects grain size that can be compensated by improving the remobilization of water-soluble carbohydrates stored in stem and leaf sheaths to developing grains under high temperature and drought stress [14]. However, studies showing the direct relationship of water-soluble carbohydrates with grain yield under heat or drought stress are limited. In addition, wheat plants with cooler canopy during grain filling, have capability to access subsoil moisture which helps to maintain evaporation and photosynthesis under hot irrigated conditions [16]. Wheat plants with cooler canopies also have positive association with grain yield [13,17,18]. Therefore, these physiological traits can be the best combination for genetic improvement of wheat genotypes to mitigate heat and drought stress because of their common genetic basis [16,19,20].

Exploring combination of physiological traits needs identification of wheat genotypes which produce high yield and express adaptation traits under high temperature exposure. Therefore, developing heat tolerant genotypes through breeding is a major objective in wheat improvement programs [3,6]. In view of global climate changes, these efforts should be accelerated to reduce high temperature effects [1,21] as rise in the global average temperature is a serious threat to wheat production in different MEs prone to terminal drought or heat [3,22,23]. 

To cope with situation, at present, efforts for evaluation of exotic germplasm developed by CIMMYT are in progress through National Agriculture Research Programs (NARPs). For this, elite nurseries of wheat germplasm are tested in MEs of South Asia for their adaptability to heat and associated traits which are later assembled into mega varieties for both improved productivity and heat tolerance [3,6,24]. Similarly, an international wheat phenotyping network has been established in South Asia for application of phenotyping techniques to accelerate selection and support breeding program for their success by incorporating physiological traits into new generation of lines for heat or drought tolerance [24]. Thus, physiological characterization of existing genotypes or plant genetic resources through phenotyping tools may provide better understanding of heat adaptive traits and their integration into breeding programs may help to translate these genotypes into desirable plant types [25,26]. The present study, therefore, compared the performance of wheat genotypes including one widely grown and two promising advanced lines with early maturing heat tolerant cultivar to identify morpho-physiological traits for adaptation to terminal heat stress under late sowing. Most specifically the association of these traits with grain yield was determined for their further use in cultivar selection. 

## 2. Results

### 2.1. Crop Phenological Development

High temperature significantly reduced crop development period, including booting (7–8 d), heading (2 d), anthesis (3 d), grain filling (10 d) and maturity period (10–12 d) under late sowing. Among the genotypes, earlier heading (3, 3, 4 d), booting (9–11, 10–12, 11–15 d), anthesis (1–2, 2–5, 4 d) and maturity (10–13, 10–12, 10–11 d) were observed for Punjab-11, V-07096 and V-10110 respectively compared to heat tolerant Millet-11 under late sowing condition. However, delayed heading (2 d), booting (3 d) and anthesis (7 d) were observed for heat tolerant Millet-11 while grain filling period was reduced for all genotypes under late sowing condition. Interactions were significant for all phenological traits (Table 1). 

### 2.2. Physiological Traits

Wheat crop expressed a 20% and 71–125% increase in Chl *a* and *b* contents respectively, under late sowing compared to timely sown crop during anthesis. Heat tolerant check Millet-11 and advanced line V-07096 expressed the highest Chl *a* and *b* contents during the first growing season (2012–2013), while these were highest in Punjab-11 and V-10110 during the second growing season (2013–2014) under late sowing. Reduced canopy temperature was observed in late sown crop during 2012–2013 and vice versa for 2013–2014. Heat tolerant Millet-11 and advanced lines V-07096 and V-10110 expressed reduced canopy temperature during both seasons. Maximum increase (50%) in stem water-soluble carbohydrates was found under late sowing. Nonetheless, all genotypes expressed similar stem water-soluble carbohydrates under late sowing (Table 1).

### 2.3. Gas Exchange Traits

No significant difference was observed for photosynthetic (A) and transpiration rates (E) under timely and late sown wheat while intercellular CO_2_ concentration (Ci) and stomatal conductance (Gs) were reduced under late sowing. However, lowest photosynthetic and transpiration rates were found for advanced line V-07096. Regarding interactions, the highest Gs was found for timely sown advanced line V-07096 and it was drastically reduced under late sowing condition. On the other hand, highest Ci was found in advanced line V-07096 under both sowing conditions while Gs was significantly reduced under late sowing (Figure 1).

### 2.4. Yield Related Traits

Terminal heat stress significantly reduced yield related traits in all the wheat genotypes under late sowing. Among these traits, plant height, spike length, grains per spike, thousand grain weight and biological yield were reduced more under late sowing than timely sowing (Table 2). Total number of productive tillers were highest during both years, while grain yield was high during first growing season and was similar under both timely and late sowing in 2nd growing season. Advanced lines V-07096 and V-10110 expressed the highest plant height, spike length, number of grains per spike, thousand grain weight including biological and grain yields. Genotype Punjab-11 also expressed highest biological and grain yields significantly similar to advanced line V-07096. For total productive tillers, the differences among genotypes were non-significant. The interactions were significant for spike length, thousand grain weight and biological yield, while these traits decreased during second growing season, grain yield during both years and plant height during first growing season were significant (Table 2). 

Positive correlation was recorded for Chl *a* and *b* contents, and canopy temperature with grain yield while no relationship was noted between stem water-soluble carbohydrates and grain yield under both timely and late sowing (Figure 2). A significantly strong, and positive relationship of 1000-grain weight was found with days to heading during both years while with maturity time during the 2nd year only. Nonetheless, the relationship of grain yield with heading and maturity time was non-significant during both years and negative with maturity time during the 1st growing season (Table 3). 

## 3. Discussion 

Identification of plant traits and developing wheat cultivars with improved adaptation to terminal heat stress is priority for plant breeders around the globe. The present study compared the performance of wheat genotypes to identify the morpho-physiological traits for terminal heat stress tolerance under late sowing. Results showed that late sowing reduced yield traits, including spike length, number of grains per spike, thousand grain weight and biological yield than timely sown crop, albeit response varied between sowing conditions and years. The reduced number of grains per spike in the present study might be due to low grain fertility and floral spikelets associated with increased temperature even by 1 °C during booting and anthesis stages [27]. Reduction in number of grains depends on developmental stage, at which high temperature occurs and determined by supply of carbohydrates during floral development, which is a sensitive process to high temperature [28]. Higher number of grains per spike in advanced lines of the present study might be attributed to increased availability of stem water-soluble carbohydrates. Advanced lines including genotype Punjab-11 had higher thousand grain weight and grain yield than heat tolerant early maturing check Millet-11 and maintained it under terminal heat stress. Better agronomic and yield performance of the advanced lines and Punjab-11 seemed to be genotypic specific effects [29], while achieving and maintaining the optimal grain weight is considered an index of heat stress tolerance and adaptation [30]. Another possible reason for the decrease in yield traits of the present study might be attributed to reduced duration between different crop stages and less translation of biomass to yield (Table 1 and Table 2). Reduced biological yield, thousand grain weight and 6–20% decrease in grain yield is reported in wheat crop exposed to terminal heat stress [3,23,31]. 

Nonetheless, a higher number of total productive tillers and grain yield during the first growing season under late sowing might be due to superiority in physiological traits of genotypes expressing high leaf chlorophyll contents and cooler canopies during anthesis or grain filling stages. Both traits help the plants to maintain better photosynthetic performance and remobilization of stem reserves to the developing grains during grain filling under high temperature or drought stress [3]. High temperature increased transpiration and keep crop canopies cool and turgid to maintain the photosynthetic performance [16]. Although relatively low transpiration and photosynthetic rates were found especially for advanced line V-07096 under late sowing, these were significantly similar with timely sowing supporting the hypothesis that genotypes maintained photosynthetic performance under terminal heat stress.

Reduced canopy temperature response of wheat lines V-07096 and V-10110 was also reflected with a decrease in gas exchange traits including A, E and Ci under late sowing in the present study [16]. Higher leaf chlorophyll contents at anthesis in wheat genotypes are an indicative of delayed senescence, high photosynthetic rate and remobilization of assimilates under terminal heat stress. Both traits contributed to higher grain yield in wheat genotypes under late sowing as evident from direct association of leaf chlorophyll contents and canopy temperature with grain yield (Figure 2a,b). Stay green trait is highly dependent on the environment and has a strong positive relationship with grain filling rate, duration and grain yield under heat stress [20]. Grain yield and stay grain are controlled by similar quantitative traits loci (QTLs) which are co-localized for productivity enhancement under heat stressed environments [20]. Heat tolerant wheat lines developed with physiological traits having cooler canopies and stay green showed superior yield and higher thousand grain weight with better adaptability under terminal heat stress [3,24]. 

The stem water-soluble carbohydrate is a potential adaptive trait for developing heat or drought tolerant wheat and 10–50% variation in stem water-soluble carbohydrates to total stem dry weight has been reported in wheat [32,33]. Higher accumulation of stem water-soluble carbohydrates in advanced lines along with early maturing heat tolerant check Millet-11 of the present study indicated increased buffering capacity of these genotypes to remobilize carbon reserves towards the developing grains accumulated during stem elongation period under terminal heat stress [14]. 

Nonetheless, no relationship of stem water-soluble carbohydrates with grain yield in the present study (Figure 2d) validates that weak association of water-soluble carbohydrates with yield as reported earlier and response is dependent on environment [16,26]. Higher stem water-soluble carbohydrates expressed under late sowing in the present study validate the potential of trait under stress condition and should be further investigated. In addition to superior physiological traits, wheat genotypes showed delayed maturity, however, response was compensated with higher yields ranging 5.38–23.82% compared to early maturing heat tolerant check Millet-11. Early maturity is as considered a breeding criterion to escape the effects of terminal heat stress, however, short duration may be accompanied with grain yield losses [3]. Interestingly, association of thousand grain weight with days to heading was significant and strong during both years and with days to maturity in the 1st year (Table 3). 

Nonetheless, superior performance of advanced lines and Punjab-11 for grain yield and physiological traits demonstrated their potential to be used for the physiological breeding programs. Thus genotypes with heat adaptive traits should be considered in parent selection for a targeted environment or for the identification of one or more adaptive traits [24,25,26,27,28,29,30,31,32,33,34,35,36,37,38,39,40,41,42,43,44,45,46]. 

## 4. Materials and Methods 

### 4.1. Study Site 

This study was conducted during the winter (rabi) growing season of 2013–2014 and 2104–2015 at University of Agriculture Faisalabad (latitude, 31° 26’ N; longitude, 73° 06′ E; altitude 184.4 m). The soil had sandy loam texture with Lyallpur series, an Aridisol, a fine silty, mixed, hyperthermic Ustalfic Haplargid according to USDA classification. Seeds of wheat genotypes were obtained from Wheat Research Institute, Ayub Agriculture Research Institute (AARI), Faisalabad.

### 4.2. Plant Material 

The four wheat genotypes used in the present study were two widely cultivated (Millet-11, Punjab-11) and two promising lines (V-07096, V-10110). 

### 4.3. Creation of Heat Stress Environment and Experimental Design

Genotypes were cultivated at two sowing times viz. timely and late sowing. The crop planted on the 10th and 13th of November was considered as normal sowing while on the 10th and 13th of December was recorded as late sowing during 2013 and 2014, respectively. The delayed sowing was done with the objective to create a heat stress environment at anthesis and during reproductive stages. The early maturing genotype Millet-11 was used as heat tolerant check while Punjab-11 was selected for its high yield potential [47]. The experimental treatments were randomized in complete block design (RCBD) with split-plot arrangement with sowing dates into main plots and wheat genotypes into sub-plots and each replicated thrice. Wheat genotypes were planted into net plot size of 5.7 m × 3.5 m at inter-row spacing of 22.5 cm. 

### 4.4. Climate and Weather Conditions

The Faisalabad features semi-arid sub-tropical climate and is located in Punjab-Pakistan. The rice-wheat rotation is generally practiced as a cropping pattern in this ecological zone. Wheat is grown as a spring crop starting from November and harvested in April. The average maximum and minimum temperatures during winter were 21 °C and 6 °C, respectively. May, June and July are the hottest months during summer and December, January and February are the coldest ones during winter [48]. Average annual rainfall is about 300 mm and that is highly seasonal and 50% of it is received during monsoon in July and August. However, in the present study, the temperature from sowing to booting ranged between 16 to 28 °C during 2012–2013 and 9.6 to 28.1 °C during 2013–2014. The maximum temperature of wheat crop sown under normal and late sowing from anthesis to maturity including grain filling period ranged from 16.8 to 31.5 °C in 2012–2013 and 17.3 to 33.6 °C in 2013–2014 of present study and similar trend was observed for low temperature (Figure 3).

### 4.5. Crop Husbandry

The seed rate of 125 kg ha^–1^ was used for wheat cultivation. Recommended fertilizers doses of nitrogen (N), phosphorus (P) and potassium (K) were applied at rate of 100–85–60 kg ha^−1^ using urea, single super phosphate and sulfate of potash, respectively. Whole of the P and K were applied during land preparation and while N was applied in three splits with the 1st half during land preparation as basal, 2nd at the 1st irrigation and 3rd at the 2nd irrigation. Crop was irrigated four times including the 1st as pre-saturated, 2nd at crown initiation, 3rd at tillering and 4th during anthesis stages. All plant protection measures were performed as recommended and weeds were controlled manually.

### 4.6. Observations

#### 4.6.1. Crop Phenology 

Phenological measurements at different crop developmental stages such as days to booting, heading, anthesis and maturity including grain filling period were recorded following Zadoks scale [49]. 

#### 4.6.2. Physiological Traits

For canopy temperature, measurements were taken during mid grain-filling stage using Infrared Thermometer between 10:00 h to 14:00 h under clear bright sky with no wind. Two readings per plot were taken and averaged. Leaf chlorophyll contents were measured according to Arnon et al. [50]. Five flag leaves were harvested from each plot at 10th days after anthesis and 0.5 g leaf sample was extracted in acetone (80% *v*/*v*) and kept overnight in sealed falcon tubes at 5 °C. The absorbance of the supernatant was determined at 645 nm and 663 nm using a spectrophotometer (Hitachi-220 Japan). Water soluble carbohydrates i.e., stem reserves were measured according to Yemm and Willis [51] and stem from main primary tillers was collected at the 7th day after anthesis. Fresh material (0.5 g) was boiled in 5 mL distilled water for 1 h. The extract was filtered and distilled water was added up to 50 mL. The 5 mL anthrone reagent was added into 1 mL of the extract along the sidewall of the test tube and vortexed. The mixture was heated in a water bath at 90–95 °C for 20 min, cooled down and absorbance was taken at 620 nm using a spectrophotometer (Hitachi-220 Japan). 

#### 4.6.3. Gas Exchange Traits 

Stomatal conductance (gs), sub-stomatal CO_2_ concentration (Ci), photosynthetic (A), and transpiration rates (E) were recorded at anthesis stage from top the 3rd leaf of every plant using infra-red gas analyzer (IRGA) (model, LCA-4; Analytical Development Company, Hoddesdon, England) [52]. All these measurements were recorded at 13.00–14.00 h. During measurements, leaf chamber molar gas flow rate was 248 µmol s^−1^, ambient CO_2_ conc. (Cref) 352 µmol mol^−1^, ambient pressure (P) of 98.01 k Pa, molar flow of air/leaf area 221.06 mol m^−2^s^−1^ and leaf chamber volume with gas flow rate (v) of 380 mL/min.

#### 4.6.4. Yield Related Traits

Average plant height, spike length and number of grains per spike were recorded at maturity by threshing 10 primary tillers manually. Numbers of productive tillers of each genotype from selected plants were counted at maturity and averaged. The grains of threshed plants for each replication were bulked separately for 1000-grains and weighed. For straw yield, an area of 2 m^2^ was randomly harvested twice per plot, tagged and weighed separately with the help of an electric balance. The harvested samples were threshed manually and grain yield for each genotype was measured. Both straw and grain yields were expressed as g m^−2^. 

#### 4.6.5. Statistical Analysis

Analysis of variance (ANOVA) was performed for data comparison using Statistix 8.1 software. The year was taken as factor to find the homogeneity or heterogeneity and where year was significant, the data of both years were analyzed and presented separately, otherwise pooled. MS-Excel was used to present data graphically. To overcome confounding effects of maturity among cultivars for physiological traits, co-variance analysis was performed. The significant difference among treatment means was compared following Tukey’s range test at 5% probability level. Pearson’s correlation was also performed for heading and maturity time with thousand grain wight and grain yield to determine relationship between them.

## 5. Conclusions

The present study revealed necessary variation among genotypes for various morpho-physiological traits associated with adaptation to terminal heat stress. Advanced lines and cv. Punjab-11 had better agronomic performance for most of the traits than heat tolerant check Millet-11 under late sowing. A positive relationship of leaf chlorophyll contents and canopy temperature was found with grain yield. Therefore, advanced lines and Punjab-11 with high biomass and grain yield, reduced canopy temperature and high leaf chlorophyll contents can be promising sources to be utilized in physiological breeding for developing climate resilience in wheat.

## Figures and Tables

**Figure 1 plants-10-00455-f001:**
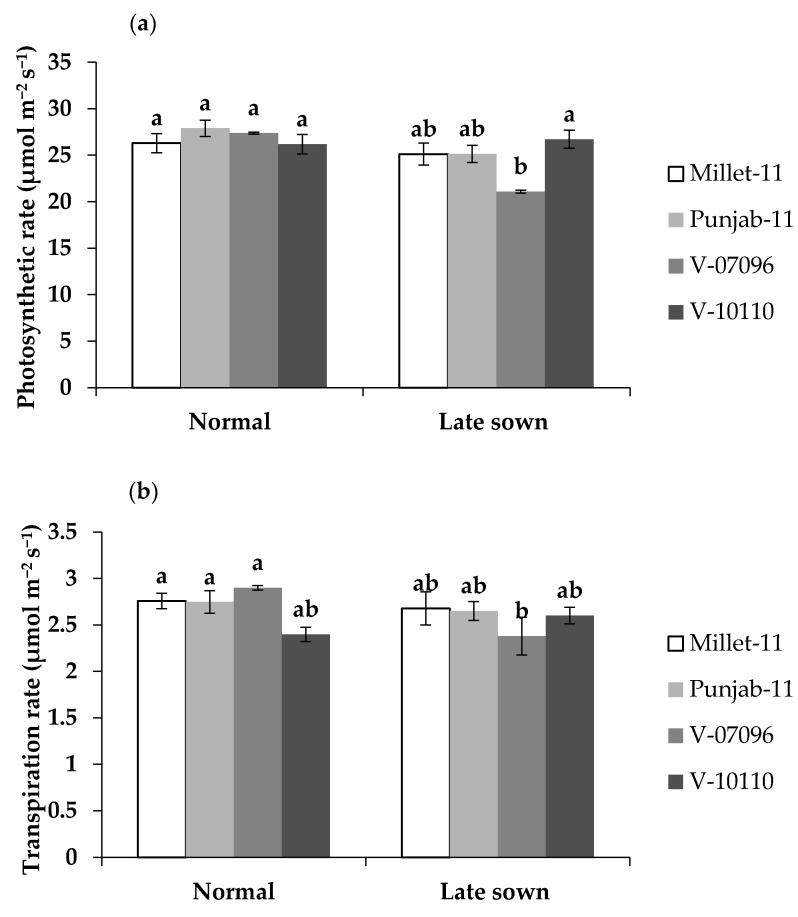
Gas exchange traits (**a**) photosynthetic rate (**b**) transpiration rate (**c**) sub-stomatal CO_2_ (**d**) stomatal conductance in wheat genotypes under late sowing. Different lowercase letters indicate significant differences among treatments; Means with same letters show non-significant difference at *p* > 0.05.

**Figure 2 plants-10-00455-f002:**
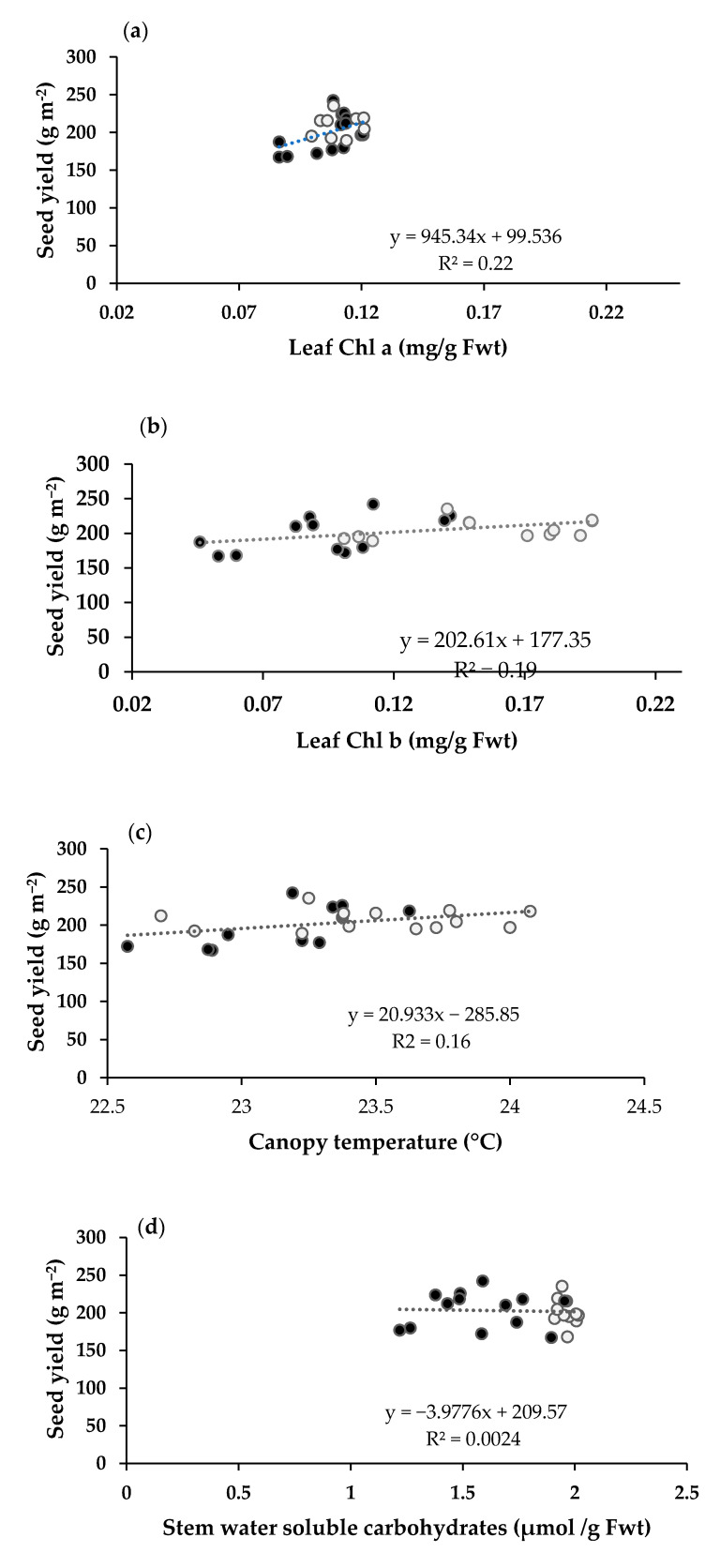
Regression relationship of seed yield with (**a**) leaf Chl a, (**b**) Chl b, (**c**) canopy temper-ature (**d**) stem water soluble carbohydrates. Filled box with black color indicate the data val-ues for timely planted and white filled for late planted wheat crop.

**Figure 3 plants-10-00455-f003:**
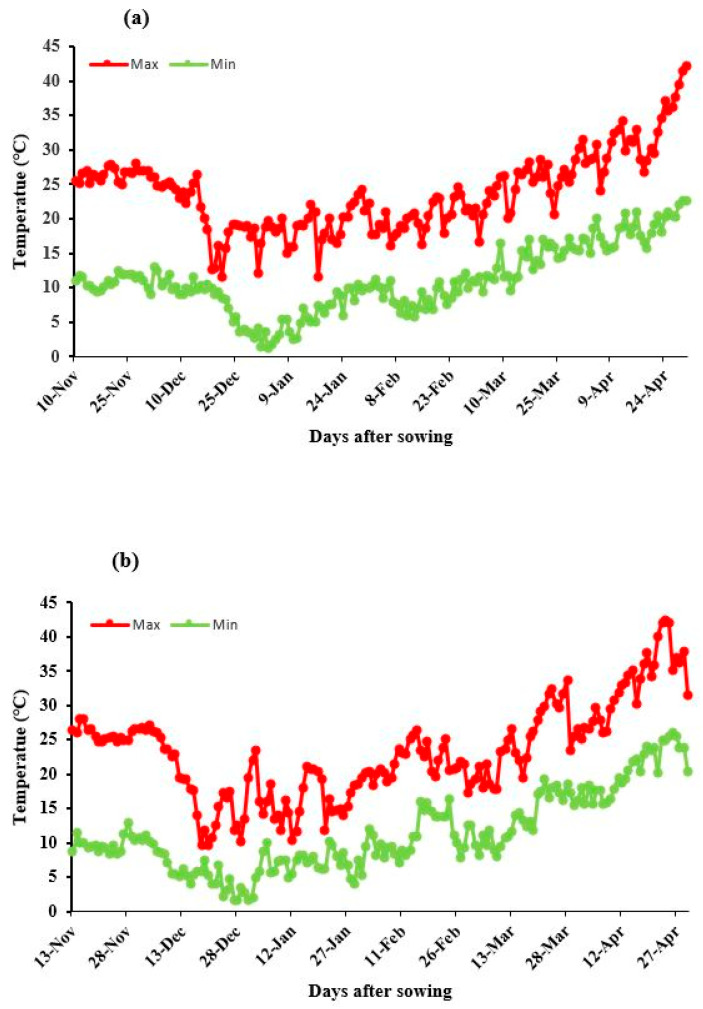
Maximum and minimum temperatures during crop growing period (**a**) 2013–2014 and (**b**) 2014–2015.

**Table 1 plants-10-00455-t001:** Mean comparison for crop phenology and physiological traits in wheat genotypes under late sown condition.

**Wheat Genotypes**	**Days to Heading**			**Grain Filling Period (Days)**		
	**2012–2013**			**2013–2014**		
	**Planting Time**		**Means** **Genotypes**	**Planting Time**		**Means** **Genotypes**
	**Normal**	**Late Sown**		**Normal**	**Late Sown**	
Millet-11	92.00 g	94.50 f	93.25 D	35.67 a	18.50 e	27.08 AB
Punjab-11	103.00 c	100.17 e	101.58 C	32.50 b	23.00 d	27.75 A
V-07096	104.83 b	101.50 d	103.17 B	30.17 c	22.67 d	26.41 B
V-10110	107.00 a	103.33 c	105.17 A	29.50 c	22.67 d	26.08 B
Means SD	101.71 A	99.87 B		31.96 A	21.71 B	
HSD	SD = 0.36; G = 0.76, SD × G = 1.30			SD = 1.08.; G = 1.00, SD × G = 1.72		
**Wheat Genotypes**	**Days to Booting**					
	**2012–2013**			**2013–2014**		
	**Planting Time**		**Means** **Genotypes**	**Planting Time**		**Means** **Genotypes**
	**Normal**	**Late Sown**		Normal	Late Sown	
Millet-11	77.67 f	80.67 e	79.17 C	78.67 f	81.33 e	80.00 C
Punjab-11	94.33 b	83.33 d	88.83 B	95.67 b	86.33 cd	91.00 B
V-07096	95.67 ab	85.33 c	90.50 A	98.33 a	86.67 c	92.50 A
V-10110	96.67 a	85.67 c	91.17 A	99.33 a	84.67 d	92.00 AB
Means SD	91.08 A	83.75 B		93.00 A	84.75 B	
HSD	SD = 0.71; G = 1.07, SD × G = 1.84			SD = 0.62; G = 1.09, SD × G = 1.88		
**Wheat Genotypes**	**Days to Anthesis**					
	**2012–2013**			**2013–2014**		
	**Normal**	**Late** **Sown**	**Means** **Genotypes**	**Normal**	**Late Sown**	**Means** **Genotypes**
Millet-11	96.67 f	103.67 e	100.17 D	99.33 d	100.67 d	100.00 D
Punjab-11	107.33 cd	106.67 d	107.00 C	111.33 b	108.33 c	109.83 C
V-07096	110.33 b	108.00 bc	109.17 B	114.33 a	109.33 bc	111.83 B
V-10110	112.00 a	108.67 bc	110.33 A	115.33 a	111.33 b	113.33 A
Means SD	106.58	106.75		110.08 A	107.42 B	
HSD	SD = n.s.; G = 0.76, SD × G = 1.30			SD = 0.36; G = 1.31, SD × G = 2.26		
**Wheat Genotypes**	**Days to Maturity**					
	**2012–2013**			**2013–2014**		
	**Normal**	**Late Sown**	**Means** **Genotypes**	**Normal**	**Late Sown**	**Means**
						**Genotypes**
Millet-11	132.67 b	122.33 c	127.50 B	134.67 b	119.00 d	126.83 D
Punjab-11	140.33 a	130.33 b	135.33 A	143.33 a	130.67 c	137.00 C
V-07096	140.67 a	130.33 b	135.50 A	144.33 a	132.33 b	138.33 B
V-10110	141.67 a	130.67 b	136.17 A	144.67 a	134.67 b	139.67 A
Means SD	138.83 A	128.42 B		141.75 A	129.17 B	
HSD	SD = 1.43; G = 0.99, SD × G = 1.71			SD = 0.95; G = 1.10, SD × G = 1.91		
**Wheat Genotypes**	**Chl *a* (mg/g Fwt)**					
	**2012–2013**			**2013–2014**		
	**Normal**	**Late Sown**	**Means**	**Normal**	**Late Sown**	**Means**
			**Genotypes**			**Genotypes**
Millet-11	0.10 d	0.12 a	0.11	0.08 e	0.09 d	0.08 C
Punjab-11	0.10 cd	0.11 b	0.10	0.12 ab	0.12 ab	0.12 A
V-07096	0.11 c	0.12 ab	0.11	0.11 c	0.09 d	0.10 B
V-10110	0.11 c	0.11 b	0.11	0.11 bc	0.13 a	0.12 A
Means SD	0.10 B	0.12 A		0.10	0.10	
HSD	SD = 0.006; G = n.s., SD × G = 0.006			SD = n.s.; G = 0.004, SD × G = 0.006		
**Wheat Genotypes**	**Chl *b* (mg/g Fwt)**					
	**2012–2013**			**2013–2014**		
	**Normal**	**Late Sown**	**Means**	**Normal**	**Late Sown**	**Means**
			**Genotypes**			**Genotypes**
Millet-11	0.07	0.15	0.11 D	0.03 e	0.03 e	0.03 D
Punjab-11	0.09	0.19	0.14 B	0.08 d	0.19 a	0.13 A
V-07096	0.10	0.19	0.15 A	0.09 c	0.09 c	0.09 C
V-10110	0.08	0.17	0.12 C	0.08 d	0.17 b	0.12 B
Means SD	0.08 B	0.18 A		0.07 B	0.12 A	
HSD	SD = 0.006; G = 0.004, SD × G = n.s.			SD = 0.006; G = 0.004, SD × G = 0.006		
**Wheat Genotypes**	**Canopy Temperature (** **°C)**					
	**2012–2013**			**2013–2014**		
	**Normal**	**Late Sown**	**Means**	**Normal**	**Late Sown**	**Means**
			**Genotypes**			**Genotypes**
Millet-11	22.53	21.32	21.92 B	23.27	25.15	24.21 AB
Punjab-11	22.67	22.36	22.52 A	23.92	25.53	24.73 A
V-07096	22.93	22.17	22.56 A	23.45	24.57	24.01 B
V-10110	22.82	22.1	22.46 AB	23.31	25.18	24.25 AB
Means SD	22.74 A	21.99 B		23.49 B	25.11 A	
HSD	SD = 0.50; G = 0.56, SD × G = n.s.			SD = 0.88; G = 0.55, SD × G = n.s.		
**Wheat Genotypes**	**Water Soluble Carbohydrates** **(μmol /g Fwt)**					
	**Average over two seasons**					
	**Normal**	**Late Sown**	**Means**			
			**Genotypes**			
Millet-11	1.90 b	2.37 a	2.14			
Punjab-11	1.34 c	1.99 ab	1.66			
V-07096	1.29 c	2.04 ab	1.67			
V-10110	1.05 d	2.04 ab	1.55			
Means SD	1.40	2.11				
HSD	SD = n.s.; G = n.s., SD × G = 0.16					

Letters among columns denote significant differences in means for sowing dates while letters within columns denote significant differences between cultivars at *P* ≤ 0.05; SD = Sowing dates; G = Genotype; n.s. = non-significant; Fwt = Fresh weight basis.

**Table 2 plants-10-00455-t002:** Mean comparison for yield related traits in wheat genotypes under late sowing condition.

**Wheat Genotypes**	**Total Productive Tillers Per Plant**					
	**2012–2013**			**2013–2014**		
	**Planting Time**			**Planting Time**		
	**Normal**	**Late Sown**	**Means** **Genotypes**	**Normal**	**Late Sown**	**Means** **Genotypes**
Millet-11	5.03	6.33	5.68	4.50	5.50	5.00
Punjab-11	5.73	6.07	5.90	5.67	6.00	5.78
V-07096	4.87	6.07	5.47	4.67	5.83	5.19
V-10110	5.13	6.20	5.67	5.44	6.67	6.06
Means SD	5.19 B	6.17A		5.01 B	6.00 A	
HSD	SD = 0.63; G = n.s. SD × G = n.s.			NS = 0.91, G = n,s, SD × G = n.s.		
**Wheat Genotypes**	**Spike Length (cm)**			**Grains Per Spike**		
	**Average over two seasons**			**Average over two seasons**		
	**Normal**	**Late Sown**	**Means** **Genotypes**	**Normal**	**Late Sown**	**Means** **Genotypes**
Millet-11	10.93 b	9.46 c	10.19 B	42.77	37.93	40.35 B
Punjab-11	10.93 b	10.02 bc	10.48 B	41.40	32.97	37.18 B
V-07096	11.91 a	10.49 b	11.20 A	48.87	44.20	46.53 A
V-10110	12.48 a	10.20 b	11.34 A	41.87	41.70	41.78 AB
Means SD	11.57 A	10.04 B		43.72	39.20	
HSD	SD = 0.61; G = 0.42, SD × G = 0.72			SD = n.s.; G = 4.79, SD × G = n.s.		
**Wheat Genotypes**	**Thousand grain weight (g)**					
	**2012–2013**			**2013–2014**		
	**Normal**	**Late Sown**	**Means** **Genotypes**	**Normal**	**Late Sown**	**Means** **Genotypes**
Millet-11	40.67	39.00	39.83 B	43.17 bc	42.00 c	42.58 B
Punjab-11	39.83	41.20	40.51 B	48.83 ab	43.33 bc	46.08 A
V-07096	47.67	47.90	47.78 A	49.50 a	39.83 c	44.67 AB
V-10110	46.00	48.23	47.11 A	51.50 a	43.17 bc	47.33 A
Means SD	43.54	44.08		48.25 A	42.08 B	
HSD	SD = n.s.; G = 3.73, SD × G = n.s.			SD = 4.40; G = 3.43, SD × G = 5.99		
**Wheat Genotypes**	**Biological Yield (g m^−2^)**					
	**2012–2013**			**2013–2014**		
	**Normal**	**Late Sown**	**Means** **Genotypes**	**Normal**	**Late Sown**	**Means** **Genotypes**
Millet-11	1212.20	1049.70	1130.90 B	1110.50 b	988.00 b	1049.30 B
Punjab-11	1448.80	1216.70	1332.80 A	1472.90 a	1053.80 a	1263.40 A
V-07096	1577.20	1247.00	1412.10 A	1462.00 a	1106.30 b	1284.20 A
V-10110	1238.30	1137.80	1188.10 B	1129.00 b	973.80 b	1051.40 B
Means SD	1369.10 A	1162.80 B		1293.60 A	1030.50 B	
HSD	SD = 79.01; G = 122.18, SD × G = n.s.			SD = 107.88; G = 83.03; SD × G = 143.1		
**Wheat Genotypes**	**Grain Yield (g m^−2^)**					
	**2012–2013**			**2013–2014**		
	**Normal**	**Late Sown**	**Means** **Genotypes**	**Normal**	**Late Sown**	**Means** **Genotypes**
Millet-11	199.18 b	228.67 ab	213.93 B	149.37 c	155.93 c	152.65 B
Punjab-11	251.87 a	239.50 ab	245.69 A	198.92 a	183.33 ab	191.13 A
V-07096	245.97 a	253.72 a	249.84 A	194.60 a	190.74 ab	192.67 A
V-10110	195.28 b	255.58 a	225.43 AB	163.24 bc	144.32 c	153.78 B
Means SD	223.08 B	244.37 A		176.53	168.58	
HSD	SD = 13.72.; G = 26.67, SD × G = 45.97			SD = n.s.; G = 12.55, SD × G = 21.63		
**Wheat Genotypes**	**Plant Height (cm)**					
	**2012–2013**			**2013–2014**		
	**Normal**	**Late sown**	**Means** **Genotypes**	**Normal**	**Late sown**	**Means** **Genotypes**
Millet-11	105.88 bc	90.13 e	98.01 B	95.07	88.47	91.77 B
Punjab-11	106.93 b	95.78 de	101.35 B	106.10	95.73	100.92 A
V-07096	115.67 a	104.83 bc	110.25 A	103.60	98.90	101.25 A
V-10110	119.50 a	100.19 cd	109.84 A	101.40	91.50	96.45 AB
Means SD	111.99 A	97.73 B		101.54 A	93.65 B	
HSD	SD = 6.37; G = 3.48, SD × G = 6.69			SD = 0.02.; G = n.s., SD × G = n.s.		

Letters among columns denote significant differences in means for sowing dates while letters within columns denote significant differences between cultivars at *P* ≤ 0.05; SD = Sowing dates; G = Genotype; n.s. = non-significant.

**Table 3 plants-10-00455-t003:** Correlation co-efficient of heading and maturity time with thousand seed weight and seed yield.

Pearson’s Correlation (*r_p_*) for Thousand Grain Weight and Grain Yield				
	Thousand Grain Weight		Grain Yield	
	1st Year	2nd Year	1st Year	2nd Year
Days to heading	0.61 (0.0016)	0.58 (0.0034)	0.23 (0.2764)	0.37 (0.0792)
Days to maturity	0.27 (0.1981)	0.81 (>0.0001)	−0.12 (0.5871)	0.39 (0.0604)

## Data Availability

Data is contained within the article.

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
