# Peer review of "Evaluation of Physiological and Morphological Traits for Improving Spring Wheat Adaptation to Terminal Heat Stress"

_plants, 2021, doi:10.3390/plants10030455_

Round 1
Reviewer 1 Report
The authors have considered several of the questions reported in previous review to improve the quality of the article, especially in the introduction and discussion. Nonetheless,
1) English in the present manuscript require a further improvement. Please consider using a language editing service. The authors should carefully check the manuscript to eliminate some grammatical errors. In some cases, combining sentences could enhanced the manuscript reading.
Here are some examples:
Line 42-44: ME1 and ME5 characterized by terminal high temperature stress is highly productive irrigated environment in South Asia for wheat production and consumption in the world [3].
Moreover, this sentence describes two MEs. In the previously version of manuscript the authors describe only wheat M1 as representative location for Punjab-Pakistan. Could you please clarify?
Line 55-57: Late sown wheat experiences higher canopy temperature (> 31°C) during reproductive period, which shortens crop growth cycle thereby reducing yields [6]. High temperature reduced growth and yields in wheat are associated with adverse effects on physiological processes.
I suggest combining these two sentences into one and subordinating where possible.
Line 48-50: . And more than 13.9 million.......
This sentence tends to be redundant. Check also “And” as connective after full stop.
2) The authors have revised the results, but additional improvement is required. The description of table 3 should be analysed more in depth. Also, the authors should provide a more exhaustive caption.
Line 160-162: Likely, relationship of days to heading and maturity with thousand seed weight was significantly positive and was also positive with seed yield of former and negative with later during 1st year but non-significant
The authors should report the correct interpretation of p-values for the correlation between crop phenology traits and yield components in the first and second year.
3) as reported for the results, thediscussion needs also further improvement, although the text has been revised.
Here is an example:
Line 167-168: Results showed that late sowing of wheat reduced most of yield traits than timely planted crop
Table 2 reports different responses to planting date and between years. What is stated in the sentence is true in some cases, but significant results need more in-depth discussion.
4) Minor points to take in account:
- Table 1 and table 2: check the format (bold or non-bold characters and line alignment)
- Table 2: HDS line check “SD=”
- Figure 1 and Figure 2: I suggest formatting in multi panels as follow:
|
(a)
|
(b) |
|
(c)
|
(d) |
and to insert the figures closer to their first citation into the text
- Line 299-300, line 303: it is necessary to add the name of first author et al. before numerical citation placed in square brackets.
Leaf chlorophyll contents were measured according to [32].
Change in
Leaf chlorophyll contents were measured according to Arnon et al. [32].
- References must be numbered in order of appearance in the text. Check the right order of appearance in the Discussion and then in Materials and methods and accordingly also the References list
Author Response
Open Review
(x) I would not like to sign my review report
( ) I would like to sign my review report
English language and style
( ) Extensive editing of English language and style required
(x) Moderate English changes required
( ) English language and style are fine/minor spell check required
( ) I don't feel qualified to judge about the English language and style
|
Yes |
Can be improved |
Must be improved |
Not applicable |
|
|
Does the introduction provide sufficient background and include all relevant references? |
(x) |
( ) |
( ) |
( ) |
|
Is the research design appropriate? |
(x) |
( ) |
( ) |
( ) |
|
Are the methods adequately described? |
(x) |
( ) |
( ) |
( ) |
|
Are the results clearly presented? |
( ) |
(x) |
( ) |
( ) |
|
Are the conclusions supported by the results? |
(x) |
( ) |
( ) |
( ) |
Comments and Suggestions for Authors
The authors have considered several of the questions reported in previous review to improve the quality of the article, especially in the introduction and discussion. Nonetheless,
1) English in the present manuscript require a further improvement. Please consider using a language editing service. The authors should carefully check the manuscript to eliminate some grammatical errors. In some cases, combining sentences could enhanced the manuscript reading.
Here are some examples:
Line 42-44: ME1 and ME5 characterized by terminal high temperature stress is highly productive irrigated environment in South Asia for wheat production and consumption in the world [3].
Ans: Corrected and incorporated
Moreover, this sentence describes two MEs. In the previously version of manuscript the authors describe only wheat M1 as representative location for Punjab-Pakistan. Could you please clarify?
Ans: One of reviewers suggested to extend it across South Asia further so according to classification in India, another environment ME5 also exposed to terminal heat.
Line 55-57: Late sown wheat experiences higher canopy temperature (> 31°C) during reproductive period, which shortens crop growth cycle thereby reducing yields [6]. High temperature reduced growth and yields in wheat are associated with adverse effects on physiological processes.
I suggest combining these two sentences into one and subordinating where possible.
Ans: Incorporated
Line 48-50: . And more than 13.9 million.......
This sentence tends to be redundant. Check also “And” as connective after full stop.
Ans: Incorporated
2) The authors have revised the results, but additional improvement is required. The description of table 3 should be analysed more in depth. Also, the authors should provide a more exhaustive caption.
Line 160-162: Likely, relationship of days to heading and maturity with thousand seed weight was significantly positive and was also positive with seed yield of former and negative with later during 1st year but non-significant.
The authors should report the correct interpretation of p-values for the correlation between crop phenology traits and yield components in the first and second year.
Ans: Corrected and incorporated
3) as reported for the results, the discussion needs also further improvement, although the text has been revised.
Here is an example:
Line 167-168: Results showed that late sowing of wheat reduced most of yield traits than timely planted crop
Table 2 reports different responses to planting date and between years. What is stated in the sentence is true in some cases, but significant results need more in-depth discussion.
Ans: Thanks, sentence revised and incorporated.
4) Minor points to take in account:
- Table 1 and table 2: check the format (bold or non-bold characters and line alignment)
- Table 2: HDS line check “SD=”
- Figure 1 and Figure 2: I suggest formatting in multi panels as follow:
|
(a)
|
(b) |
|
(c)
|
(d) |
and to insert the figures closer to their first citation into the text
- Line 299-300, line 303: it is necessary to add the name of first author et al. before numerical citation placed in square brackets.
Leaf chlorophyll contents were measured according to [32].
Change in
Leaf chlorophyll contents were measured according to Arnon et al. [32].
- References must be numbered in order of appearance in the text. Check the right order of appearance in the Discussion and then in Materials and methods and accordingly also the References list
Ans: All suggested changes have been incorporated.

Reviewer 2 Report
The authors improved the paper according to the comments.
But, still necessary to make some corrections:
1) Line 119. Change 2.2. to 2.3.
2) Line 142. Change 2.5. to 2.4.
3) Figure 2 - use the bold font. Also, the charts should be presented more clearly, for example using a black font color (not gray). Check, please.
4) Figure 3 - Use the bold font and use the dot instead of the colon (Figure 3. instead of Figure 3:). Also, check the size fonts in charts, they should be the same as in other figures (use the same style of fonts and size in all presented figures).
Best regards,
Reviewer 2.
Author Response
Open Review
(x) I would not like to sign my review report
( ) I would like to sign my review report
English language and style
( ) Extensive editing of English language and style required
( ) Moderate English changes required
( ) English language and style are fine/minor spell check required
(x) I don't feel qualified to judge about the English language and style
|
Yes |
Can be improved |
Must be improved |
Not applicable |
|
|
Does the introduction provide sufficient background and include all relevant references? |
(x) |
( ) |
( ) |
( ) |
|
Is the research design appropriate? |
(x) |
( ) |
( ) |
( ) |
|
Are the methods adequately described? |
(x) |
( ) |
( ) |
( ) |
|
Are the results clearly presented? |
( ) |
(x) |
( ) |
( ) |
|
Are the conclusions supported by the results? |
(x) |
( ) |
( ) |
( ) |
Comments and Suggestions for Authors
The authors improved the paper according to the comments.
Ans: Thank you
But, still necessary to make some corrections:
1) Line 119. Change 2.2. to 2.3.
Ans: Incorporated
2) Line 142. Change 2.5. to 2.4.
Ans: Incorporated
3) Figure 2 - use the bold font. Also, the charts should be presented more clearly, for example using a black font color (not gray). Check, please.
Ans: Incorporated
4) Figure 3 - Use the bold font and use the dot instead of the colon (Figure 3. instead of Figure 3:). Also, check the size fonts in charts, they should be the same as in other figures (use the same style of fonts and size in all presented figures).
Ans: Incorporated

Reviewer 3 Report
This is my second review of this manuscript. The authors have made some revisions according to some of my earlier suggestions. I still believe the standard of English is not good and could be further improved.
Author Response
Open Review
(x) I would not like to sign my review report
( ) I would like to sign my review report
English language and style
(x) Extensive editing of English language and style required
( ) Moderate English changes required
( ) English language and style are fine/minor spell check required
( ) I don't feel qualified to judge about the English language and style
|
Yes |
Can be improved |
Must be improved |
Not applicable |
|
|
Does the introduction provide sufficient background and include all relevant references? |
(x) |
( ) |
( ) |
( ) |
|
Is the research design appropriate? |
(x) |
( ) |
( ) |
( ) |
|
Are the methods adequately described? |
(x) |
( ) |
( ) |
( ) |
|
Are the results clearly presented? |
( ) |
(x) |
( ) |
( ) |
|
Are the conclusions supported by the results? |
(x) |
( ) |
( ) |
( ) |
Comments and Suggestions for Authors
This is my second review of this manuscript. The authors have made some revisions according to some of my earlier suggestions. I still believe the standard of English is not good and could be further improved.
Ans: Thank you, we have revised the manuscript and improved even for language.

This manuscript is a resubmission of an earlier submission. The following is a list of the peer review reports and author responses from that submission.
Round 1
Reviewer 1 Report
In this manuscript the authors propose a physio-morphological characterization of wheat genotypes for better crop performance under heat stress events.
The authors evaluate the performance by comparing two advanced spring wheat lines with two spring wheat check cultivars widely grown. The authors should better specify the nature of this material (i.e. promising heat tolerant lines, known checks). Millet-11 is described as early maturing cultivar (i.e. in line 26, line 252) and then used as heat tolerant check, but not information for Punjab-11 cultivar as reported in the abstract and materials and methods.
I would recommend the authors to reorganize the introduction to make the reading more fluids and bring the reader to into the context of the problem at the basis of the study and its aim. Generally, some arguments are expressed several times on the text or overlap in some ways and more literature is needed to discuss impact of climate change on global wheat cropping system and the decline of yield.
Particularly, I suggest to link for example the sentence from line 81 to line 83 with the first part at the beginning of the introduction (from line 44 to line 51) highlighting a possible higher incidence of increasing temperatures on wheat yield in cropping areas of South Asia compared to others in the world. In addition, I suggest to better link the info on wheat cultivation in South Asia (crop rotation, growing season)
Here a list of some sentences that need to be addressed in the introduction:
Line 3: instead of “…adaptation of wheat to terminal heat stress”
You might write
adaptation of spring wheat to terminal heat stress
or
adaptation of wheat (Triticm aestivum L.) to terminal heat stress
Line 46: instead of ”...and ME1 is highly productive…..”
should be better
…and ME1, characterized by terminal high temperature stress, is highly productive….
Line 52-53: “Spring wheat in South Asia including Pakistan is grown from November-December while…”
should be expressed more clearly
Line 86-89: revise writing and position in the text
Line 90: citation number 5 regards cotton-wheat zone. the sentence could be change in "in rice-wheat and cotton wheat zones…..”
the numerical sequence of citations, placed in square brackets, should be changed as follow
Line 70: [15, 11, 12] change in [11,12,15]
The manuscript shows confused description of materials and methods and needs some clarifications and precision. This paragraph needs an improved organization for plant materials, experimental design, field trials site description, meteorological information, data collection and statistical analysis
Line 238: the number of the subparagraph 4.1 is missing
Line 240: “The experimental soil….” Should be “ the soil belongs……
The subparagraphs 4.1 and 4.2 can be combined and reorganized.
Line 267: this sentence has be moved in first subparagraph when plant materials is described
Subparagraph 4.3: I suggest representing the thermopluviometric data of the experimental site for each month and year in a graphic
Line 279: Zadoks
Line 285: “method described by (31)” it is necessary to add the name of first author and et al. before numerical citation placed in square brackets. The authors have to check in all the text
Subparagraph 4.45: Statistical analysis needs to be improved in the application and description.
The results are presented in a logical order but needs to be improve the description and linked tables
It would be nice if the authors consider expand the results and showing them one by one with more details especially for crop phenological development. Table and figures should be inserted into the main text close to their first citation, as reported in the Manuscript Preparation of Plants journal. Specifically, it is necessary to report the results of ANOVA analysis in a separate table and the means comparison in 3 different tables (Crop phenology traits, physiological traits and yield traits). The caption should describe the data shown and I suggest to insert for example “Means comparison for crop phenological traits……” In this way the tables are more readable and can be reported in the text more precisely. The proper format and presentation of figures and tables in a manuscript allows a clear reading of the results.
Table 3 are not described in the results but in the discussion Line 231. A description of Pearson correlation should be report in the results and then discussed. No description of this analysis is mentioned in the materials and methods.
Some sentences reported in the discussion do not reflect the data in the tables.
I suggest to the authors for example to rewrite from line 191 to line 204, and from line 209 to line 211.
Figure1: “Heat” should be change in “Late sowing” and improve the caption adding more information of statistical analysis (i.e. Different lowercase letters indicate significant differences among ……... Means with same letters show non-significant difference at p < 0.05).
Figure 2: it should be better to distinguish with different color the points for each sowing data. Improve the caption
References: include a DOI if available.
Reviewer 2 Report
Minor revision
The title of the paper probably should be re-phrased.
Lines 6-9. It is enough to indicate the same affiliation once
Line 20. Check fonts
Line 27 Check
Line 33. Delete (WSC)
Lines 50, 60, 61, 65, 70 etc. Specify reference numbers without spaces and indicate the reference numbers in ascending order. Check throughout the text.
Figure 2. Check fonts and legends presentation, they should be the same everywhere
Revise the name of Figure 2 is not very clear
Line 238. Indicate as 4.1. Experimental Design
Line 239. Check, please
Line 277. Change 4.4.1. to 4.5.1. (Lines 281, 290, 303, 311... the same corrections are needed)
Line 285. Change (31) to [31]
Lines 328-334. Check the fonts.

Reviewer 3 Report
Review of manuscript ID: plants-1031886. ”Physio-Morphological Trait-Based Approach for adaptation of wheat to terminal heat stress” by Hafeez ur Rehman et al.
The manuscript compares several genotypes under normal and terminal heat conditions and indicates that several new lines might be useful breeding materials for those conditions. While the findings are worthy of publication, the presentation requires considerable improvement. Following are some general and specific criticisms for the authors to consider and address:
- The English in the presentation overall is poor and would benefit from many editorial and structural corrections. This makes it more difficult to assess the quality of the work, and diminishes clear description, aims, and outcomes of the study performed.
- The specific aims are not really clearly stated in Introduction.
- There are some generalizations made in the Results, where conclusive statements are made, which only apply to one of the study seasons. This gives potentially misleading conclusion, for example, L151: “Highest ……. and seed yield were expressed under late sowing condition”. This is only true for 2012-13 season.
- It is not clear at what stage the gas exchange measurements were made. The measurement temperatures reported appear very high and significantly higher than the average growth temperatures, which might be problematic in interpretation.
- I found the Discussion to be rather muddled and difficult to read. This is due to inadequate attention to sentence and paragraph structure. Each topic discussed should be treated separately indicating what was found, how this relates to other work, and conclusions, speculations, or further work. The present Discussion is not suitable for publication.
L248-252. The section describing the treatment and lines should be briefly summarized at the end of the Introduction, or at the start of the Results, to outline to the reader the experimental strategy employed.
L114-119. Gas exchange parameters as described in the text are not reflected in the Fig 1. For example, photosynthetic rates under heat treatment are not significantly different, as stated, for three of the lines, and only for V07096 appears different. Similarly for transpiration rate.
Line 150. Wrong table is referenced.
Line 43…future gains in yields must be increased by 60%. Should read “…to 60%...”
L 81-83. This is repetitive and should be omitted. Already stated earlier (and later).
L 73. I believe it is not true to say that “..evidence explaining role of water-soluble carbohydrates in determining variation among wheat genotypes for heat tolerance are limited”, as this has been demonstrated in a number of studies and is well recognized to be important under hot, dry conditions.